# Escalation-related decision making in acute deterioration: a retrospective case note review

Natasha Campling,[1,2] Amanda Cummings,[1,2] Michelle Myall,[1,2] Susi Lund,[1,2] Carl R May,[1,2,3] Neil W Pearce,[3] Alison Richardson[1,3,2]

¹School of Health Sciences, University of Southampton, Southampton, UK
²NIHR CLAHRC Wessex, University of Southampton, Southampton, UK
³University Hospital Southampton NHS Foundation Trust, Southampton, UK

**Correspondence to**
Dr Natasha Campling;
N.C.Campling@soton.ac.uk

## ABSTRACT

**Aim** To describe how decision making inter-relates with the sequence of events in individuals who die during admission and identify situations where formal treatment escalation plans (TEPs) may have utility.

**Design and methods** A retrospective case note review using stratified sampling. Two data analysis methods were applied concurrently: directed content analysis and care management process mapping via annotated timelines for each case. Analysis was followed by expert clinician review (n=7), contributing to data interpretation.

**Sample** 45 cases, age range 38–96 years, 23 females and 22 males. Length of admission ranged from <24 hours to 97 days.

**Results** Process mapping led to a typology of care management, encompassing four trajectories: early de-escalation due to catastrophic event; treatment with curative intent throughout; treatment with curative intent until significant point; and early treatment limits set. Directed content analysis revealed a number of contextual issues influencing decision making. Three categories were identified: multiple clinician involvement, family involvement and lack of planning clarity; all framed by clinical complexity and uncertainty.

**Conclusions** The review highlighted the complex care management and related decision-making processes for individuals who face acute deterioration. These processes involved multiple clinicians, from numerous specialities, often within hierarchical teams. The review identified the need for visible and clear management plans, in spite of the frame of clinical uncertainty. Formal TEPs can be used to convey such a set of plans. Opportunities need to be created for patients and their families to request TEPs, in consultation with the clinicians who know them best, outside of the traumatic circumstances of acute deterioration.

## Strengths and limitations of this study

► There is a lack of description of escalation-related decision making in the context of deterioration outside of the critical care environment. Our study setting was the comprehensive hospital environment and individuals who were facing acute deterioration that led to death.

► The study explored clinical decision-making processes: the types and range of decisions made, the involvement of families in these processes and the interaction between clinical teams. Care management trajectories provoked by acute deterioration were characterised via typology, including points of significance in the sequence of events. Contextual issues influencing decision making were described: multiple clinician involvement, family involvement and lack of planning clarity; all framed by clinical complexity and uncertainty.

► While the sample was stratified, it was small, selected from a single acute hospital trust. However, two data analysis methods were applied concurrently (followed by expert clinician review): directed content analysis and care management process mapping via annotated timelines.

► Examination of decision-making processes highlighted areas for improvement and the potential impact of formal treatment escalation plans through pre-emptive decision making and patient involvement outside of crisis situations.

and decisions because of the acuity of their condition.[1]

Previous research has focused on the illness trajectories of deteriorating patients or on clinical decision making in the specific context of critical care. The wider context of care management and related decision making remains an unresearched area. Murray *et al*[2] highlighted the value of awareness of illness trajectories as a mechanism for clinicians to help plan care to meet patients' needs and for families to cope. More recently, Etkind and colleagues[3] defined trajectories of final illness among patients who died while inpatients. These were defined a priori to

## BACKGROUND

Clinical decision making in the context of acute deterioration during hospital admission is complex. Such decisions are frequently made in the face of uncertainty, characterised by: lack of underpinning information or diagnostic clarity, necessity for rapid decision making and the inability of patients to collaborate in discussions

their case note review as: predictable (gradual deterioration during admission); predictable (rapid deterioration during admissions); unpredictable course during hospital admission; and sudden death. One hundred and forty-nine cases were examined (all deaths over 11 months on five inpatient wards where the AMBER care bundle was implemented) and characterised according to one of four trajectories.

Our study progresses the above, which focused exclusively on illness trajectories by expanding the focus to care management and understanding of the associated decision-making processes to inform clinical practice. Higginson and colleagues[4] explored this area by examining patterns of decision making, but their work was specific to critical care. Only 16 cases were examined (in combination with interviews and non-participant observation), and four trajectories with different patterns of clinical decision making were identified: curative care from admission (to critical care); oscillating curative and comfort care; shift to comfort care; and comfort care from admission. They emphasised that 'conflict' in decision making could occur between relatives and staff and between and within clinical teams.

Given the complexity of clinical decision making related to acuity and deterioration, and perhaps the likelihood for 'conflict',[4] there has been a move nationally and internationally to develop and implement formal treatment escalation plans (TEPs). Structured, procedure-specific TEPs are proposed as a mechanism by which to improve understanding and communication when escalation-related decisions need to be made and acted on.[5] They provide a framework on which to base a conversation and document treatment options that are appropriate if a patient were to become acutely unwell. They vary in both design and use.[5–7] Notable examples include: Universal Form of Treatment Options,[8] Deciding Right (http://www.nescn.nhs.uk/common-themes/deciding-right/) and Physician Orders for Life Sustaining Treatment.[9] In the UK, there is growing interest in the national initiative, instigated in 2014, led by the Resuscitation Council and the Royal College of Nursing, which generated the Recommended Summary Plan for Emergency Care and Treatment (http://www.respectprocess.org.uk/).

Despite this increasing awareness, little work has examined the implementation of TEPs. This study is part of a wider programme of work to inform the implementation and evaluation of TEPs as part of the *Complexity, Patient Experience and Organisational Behaviour* theme of the National Institute for Health Research Collaboration for Applied Health Research and Care Wessex. Additionally, the team has undertaken a review of communication and decision-making interventions directed at goals of care via a theory-led scoping review.[10]

## AIMS/OBJECTIVES

The study explored the care management of those who deteriorate and die during hospital admission, characterising the resources mobilised, in as much detail as could be tracked through recourse to case notes. The aims were: to describe how decision-making processes inter-relate with the sequence of events for individuals who die during inpatient admission and to identify situations where treatment escalation plans may have had utility.

The objectives were:

1. To identify and characterise, via the generation of a typology, the care management trajectories of hospital inpatients facing acute pathophysiological deterioration that ultimately leads to death.
2. To map clinical decision-making processes, including the involvement of patients and families in decisions, identifying what leads to and triggers changes in management.
3. To identify the potential role of treatment escalation plans in providing a framework to support discussions and recording of decisions.

## METHODOLOGY AND METHODS

### Study design

A retrospective case note review exploring the care management of those who die during hospital admission.

### Sampling strategy

The case note review followed an initial audit of death certificate review forms (DCRFs) from all deaths at a single acute hospital trust in England (n=911) within a 6-month period (January–July 2015). Case notes of a 5% sample (45 sets of notes) of patients, aged over 18 years were reviewed. The DCRF data enabled stratified sampling, ensuring appropriate representation across groups. Thirty-two mutually exclusive strata were created based on whether cases had all possible combinations of the following: do not attempt cardiopulmonary resuscitation (DNACPR), palliative care team involvement, intensive care/high dependency management, evidence of escalation/de-escalation decision and unpredictable illness trajectory.[3] Proportionate allocation was used to sample the same fraction from each strata, with a check that the total sample size was calculated as expected (ie, not affected by rounding of the numbers for each strata to integers).

### Data collection

Data were collected from case notes only. Data collection tracked the period from admission to hospital through to each patient's death to identify: (1) when decisions to escalate or de-escalate treatment were made, (2) how those decisions were made and (3) who was involved in these decisions. For those with a prolonged admission (>30 days), data collection was limited to the last 30 days of admission (but included social and clinical data regarding their admission). The following data were extracted:

- Clinical and demographic information regarding admission to hospital, including but not limited to, comorbidities and admitting specialism.
- End-of-life care (EOLC) and DNACPR information, including but not limited to, whether CPR was attempted.
- Nature of any events leading to a discussion or decision regarding levels of care, who recognised and responded to the event, actions taken, further detail on escalation or de-escalation of care and outcomes from this. Here, 'event' referred to episodes such as clinical deterioration, ward rounds, specialist review or emergence of new clinical findings.
- How decisions were documented, including clarity of documentation and use of care plans.
- Evidence of patient and/or family involvement in decision making and how patient preferences and those of others are taken into account, including whether patient wishes were known in advance.
- Ward movements.
- Date and cause of death.

Data extraction was undertaken by two clinically qualified researchers (NC and AC) and data recorded using an Excel spreadsheet pro forma (see online supplementary file 1). The pro forma was piloted on a set of notes and based on this changes to the form were made to facilitate usability and increase reliability of data extraction. This resulted in: inclusion of all causes of death (not just cause 1a but also underlying causes 1b and 1c), and enabling of free-text entry for avoidable EOLC admission and failed EOLC discharge. The revised pro forma was tested by NC and AC on an initial sample of case notes (n=8) to assess utility and consistency of data entry. No further changes were required. At the end of data collection, a process of cross-checking by both researchers helped to mitigate against errors, ensure accuracy and consistency.

### Data analysis

Two methods of data analysis were applied concurrently. First, case notes were treated as qualitative data and analysed using directed content analysis.[11] The data within the pro forma were analysed using this method and directed towards: the event leading to the decision or discussion and the action taken and resulting outcomes and details regarding involvement and discussion with the patient and family. Data comprised verbatim transcription of relevant entries in the case notes to the pro forma. Additionally, field notes were analysed to capture limitations of case notes as a data source and recurrent issues (sequence of events and triggers for decision making) across cases.

Second, care management process mapping via annotated timelines involving key events were developed for each case.[12] These timelines included: escalation and de-escalation related decisions; involvement of patient and family in decision-making; clinical treatment plans made; investigations undertaken and treatment received; and key clinical information to inform probability of outcomes and prompt decisions.

Timelines were drawn for each case (NC and AC) and then grouped independently by NC and AC (double screened) into one of four care management trajectories, which became apparent during analysis. Categorisation of cases by the researchers were compared, with input from two additional clinical members of the research team (SL and AR). Where there was initial disagreement, the pro formas were revisited in a team discussion to agree final categorisation (n=11).

Diagrams were subsequently drawn to represent the group experience of the four care management trajectories. These were iteratively refined (NC, AC, SL and AR). They were combined with tabulated data representing the cases within each trajectory and a case exemplar (case study) and sent to a group of expert clinicians (representing a wide range of specialities) for review. They were asked to consider:

- Do the four care management trajectories capture the sequence of events and decision-making processes involved?
- Do the trajectories apply to patients you have seen recently who have then gone on to die while in hospital? Could you consider how they do or do not apply?
- Do these data demonstrate potential triggers for decision making or treatment escalation planning that you would like to see put into practice?
- Is there anything in the data you are surprised by or any other comments you would like to make?

Out of 13 experts approached, seven commented in detail on the data either face-to-face or via telephone/email. Their feedback verified: the care management trajectories reflected what clinicians encounter in practice and were described in a way they could identify with; the classification of cases to the trajectories; and the authenticity of the case exemplars. They contributed to the overall interpretation of data.

### Ethical and research governance considerations

As access to patient identifiable data (case notes) was required without consent, support under section 251 of the National Health Service Act (2006) was sought and obtained via the Health Research Authority's Confidentiality Advisory Group.[13]

### Patient and public involvement (PPI)

A PPI champion worked closely with the research team on this study, and the wider programme, informing all study processes. Involvement led to the recommendation that the team solely access paper-based notes (to restrict the amount of data accessed) and not electronic medical records as originally planned.

### RESULTS

### Sample characteristics

The age range of patients included (in the review of the 45 sets of notes) was 38–96 years, with 23 female and 22

male. The length of admission ranged from <24 hours to 97 days. Thirty-five patients had a DNACPR in place at time of death. Fifteen patients had palliative care team involvement.

## A typology of care management

Analysis via process mapping led to the development of a typology of care management, encompassing four distinct trajectories. The trajectories characterised the sequence of events and decision-making processes through acute pathophysiological deterioration leading to death. They were:

1. Early de-escalation (within 24–48 hours of admission) due to catastrophic event — clinically observable signs and symptoms ± observable on imaging.
2. Treatment with curative intent throughout (no de-escalation) ± cardiopulmonary resuscitation.
3. Treatment with curative intent until significant point.
4. Early treatment limits set (within 48 hours of admission).

Table 1 displays the key characteristics of the cases represented by each trajectory. Each care management trajectory is described in sequence below, including a diagrammatic representation of the respective trajectory. Exemplar case studies for each trajectory are included in online supplementary file 2. The process of reviewing the data with expert clinicians added a valuable dimension to data interpretation. The depth and range of their feedback, via their experiential knowledge, is summarised in online supplementary file 3.

### Early de-escalation due to catastrophic event

This trajectory was characterised by hospital admission due to 'catastrophic' events (figure 1). The event had occurred outside of hospital, was evident at the point of admission and referenced individuals who were in danger of dying on admission (eg, patients who were moribund secondary to shock) or those admitted with severe, critical illnesses (eg, major cerebrovascular accidents).

Following admission, there was a period of initial escalation, with accompanying imaging, diagnostic investigations such as blood tests or ECGs and treatment with intravenous antibiotics or fluids. This escalation also encompassed senior or specialist (eg, surgical and intensive care) review.

A key feature of this trajectory was the early (within 24–48 hours) recognition of an unsurvivable or irreversible event. All cases had at least one factor that identified this including: imaging results, clinically observable diagnoses, reduced level of consciousness and/or consultant review. Following recognition of futility, discussions with family and next of kin preceded palliation in all cases bar one. In this case, deterioration and death were so rapid as to prevent timely palliation. This trajectory was generally defined by short admissions, on average, patients died within 3 days.

### Treatment with curative intent throughout

Trajectory 2 was characterised by treatment with curative intent for the duration of hospital admission (figure 2). Individuals were admitted with a variety of diagnoses, and admissions were characterised by ongoing care at ward or high dependency unit/intensive care unit (HDU/ICU) level for multiple issues. These included fluid balance management associated with cardiorenal failure or acute kidney injury, treatment of infections and management of ischaemic or arrhythmic cardiac disease.

This trajectory was also characterised by the development of new diagnoses (eg, sepsis) or sudden, unpredictable events (eg, pulmonary embolism), which ultimately led to death. However, in these cases, such events did not trigger de-escalation (as in trajectory 3); patients were actively treated until death. In five of the eight cases, unsuccessful cardiopulmonary resuscitation occurred prior to death. In the three remaining individuals, DNACPR orders had been stimulated by senior clinician reviews and/or family discussions.

The reasons underlying a lack of de-escalation related to patient characteristics, individual preferences and the delivery or focus of healthcare. Some individuals were younger or normally fit and well with minimal comorbidities, while others expressed a preference for active treatment. For some, a recent intervention with curative intent, or the fact that they were awaiting discharge or transfer to alternative settings, meant that de-escalation was not a consideration. In others, the involvement of multiple specialist teams meant that the leading specialism (and thus the team who might be expected to make de-escalation decisions) was not clear.

### Treatment with curative intent until significant point

Trajectory 3 was characterised by curative intent treatment until a significant point, triggering de-escalation of care (figure 3). These triggers included significant deterioration in the patient's current condition (in the absence of a new diagnosis), for example, a reduction in consciousness level or patient agitation/distress, and new diagnoses (eg, infection or malignancy), which led to deterioration in the patient's condition. A third trigger involved a new clinical specialism or out-of-hours review recognising poor prognosis and the futility of current treatment, prompting de-escalation.

All triggers for de-escalation prompted discussions with next of kin, family and the patient or other clinical teams (if under shared-care management). Following these discussions, multistaged de-escalation ensued. The first stage involved the setting of ceilings of care and DNACPR orders. This first stage at times occurred prior to family discussion, but such discussion always preceded the second stage, which included stopping vital sign observations, early-warning activation scores and invasive investigations/treatments. In some cases, a time and intensity limited trial of treatment (eg, antibiotics) preceded a third stage of de-escalation, palliation. For patients receiving HDU/ICU level care, the latter stages of de-escalation

**Table 1** Key characteristics of cases within the trajectories

| Trajectory 1 cases (n=10) | | | Trajectory 2 cases (n=8) | | |
|---|---|---|---|---|---|
| Age (median, range) | 79.5 (47–94) years | | Age (median, range) | 83 (53–96) years | |
| Gender | 6 female; 4 male | | Gender | 1 female; 7 male | |
| CACI* Comorbidity Score (median, range) | 5.5 (4–10) | | CACI Comorbidity Score (median, range) | 6.5 (2–9) | |
| Social history | Care/nursing home resident or respite | 2 | Social history | Care/nursing home resident or respite | 1 |
| | Home carers | 1 | | Home carers | 2 |
| Length of admission (median, range) | 3 (1–16†) days | | Length of admission (median, range) | 7.5 (2–19) days | |
| Primary reason for admission | Cerebrovascular accident | 3 | Primary reason for admission | Respiratory (infective) | 3 |
| | Gastrointestinal | 2 | | Ischaemic/arrhythmic cardiac disease | 2 |
| | Sepsis | 3 | | Fall | 1 |
| | Ischaemic cardiac disease | 1 | | Fracture | 1 |
| | Peripheral vascular disease | 1 | | Cellulitis | 1 |
| Triggers for recognition of irreversibility/ unsurvivable event Some cases has more than one trigger* | Imaging results | 3 | Ongoing care management/ treatment issues All cases had multiple issues* | Fluid balance (cardiorenal failure) | 4 |
| | Clinically observable diagnosis | 4 | | Acute (on chronic) kidney injury | 2 |
| | Consultant review | 1 | | Ischaemic/arrhythmic cardiac disease | 5 |
| | Reduced consciousness | 3 | | Respiratory tract infection | 8 |
| | | | | Urinary tract infection | 1 |
| | | | | Diabetic control | 2 |
| | | | | Pulmonary embolism | 1 |
| | | | | Respiratory failure | 4 |
| Received CCO/ITU‡ review | | 1 | Received CCO/ITU review | | 1 |
| Received HDU/ITU§ care | Intensive care unit | 1 | Received HDU/ITU care | HDU | 1 |
| | | | | ICU | 1 |
| | | | CPR¶ attempted and unsuccessful | | 5 |
| | | | Reasons for no de-escalation Some cases had more than one reason* | Awaiting transfer/discharge | 3 |
| | | | | Patient preference/limited or no family involvement | 2 |
| | | | | Young/normally fit and well/few comorbidities | 3 |
| | | | | Post (curative intent) intervention | 1 |
| | | | | Input from multiple specialist teams | 2 |
| **Trajectory 3 cases (n=18)** | | | **Trajectory 4 cases (n=9)** | | |
| Age (median, range) | 80.5 (38-88) years | | Age (median, range) | 86 (63–91) years | |
| Gender | 12 female; 6 male | | Gender | 4 female; 5 male | |
| CACI Comorbidity Score (median, range) | 6 (1-12) | | CACI Comorbidity Score (median, range) | 8 (5–14) | |
| Social history | Care/nursing home resident or respite | 1 | Social history | Care/nursing home resident or respite | 5 |

**Table 1** Continued

| Trajectory 3 cases (n=18) | | | Trajectory 4 cases (n=9) | | |
|---|---|---|---|---|---|
| | Home carers | 4 | | Home carers | 1 |
| Length of admission (median, range) | 17.5 (3-97) days | | Length of admission (median, range) | 12 (2–28) days | |
| Primary reason for admission | Gastrointestinal | 3 | Primary reason for admission | Sepsis | 3 |
| | Cerebrovascular accident | 2 | | Respiratory tract infection | 2 |
| | Respiratory tract infection | 5 | | Malignancy | 1 |
| | Urinary tract infection | 1 | | Cerebrovascular accident | 1 |
| | Specialist treatment (chemotherapy, cardio ablation) | 2 | | Fall | 1 |
| | Haematological | 1 | | Acute heart failure | 1 |
| | Fracture | 1 | | | |
| | General decline + hypertension | 2 | | | |
| | Respiratory | 1 | | | |
| Received CCO/ITU review | | 4 | Pre-existing factors for characterising admission Some cases had more than one factor* | Frailty | 5 |
| | | | | History of recent deterioration | 2 |
| | | | | Pre-existing DNACPR** | 4 |
| | | | | Current malignancy | 3 |
| | | | | Underlying dementia | 3 |
| | | | | Already known to palliative care | 3 |
| Received HDU/ITU care | Intensive care unit | 3 | Prompts for setting early treatment limits Some cases had more than one prompt* | Senior clinician review | 7 |
| | | | | Patient's prior wishes expressed by family | 4 |
| | | | | Discussion with patient | 3 |
| | | | | Marked deterioration | 4 |
| Significant point triggering de-escalation Some cases has more than one more trigger* | Significant deterioration in current condition | 6 | | | |
| | New diagnosis leading to deterioration in condition | 11 | | | |
| | New clinical team/out of hours input recognising poor prognosis | 4 | | | |

*Charlson Age Comorbidity Index (CACI; www.pmidcalc.org/7722560) (Charlson et al[20]).
†One individual lived for 16 days despite catastrophic event due to younger age.
‡Critical care outreach/intensive care review.
§High dependency/intensive care.
¶Cardiopulmonary resuscitation.
**Do not attempt cardiopulmonary resuscitation.
HDU, high dependency unit; ICU, intensive care unit.

involved the withdrawal of treatment. There was usually some degree of treatment provided in parallel to multi-staged de-escalation, although this was limited, typically involving antibiotics and intravenous fluids. The time between the significant point, which triggered de-escalation and patient death, was between 0–10 days; however, this trajectory was characterised by the longest and most varied admission length, 3–97 days.

### Early treatment limits set
Trajectory 4 was characterised by the presence of early treatment limits, set within 48 hours of admission

CARE MANAGEMENT TRAJECTORY TYPE 1

**EARLY DE-ESCALATION DUE TO CATASTROPHIC EVENT**

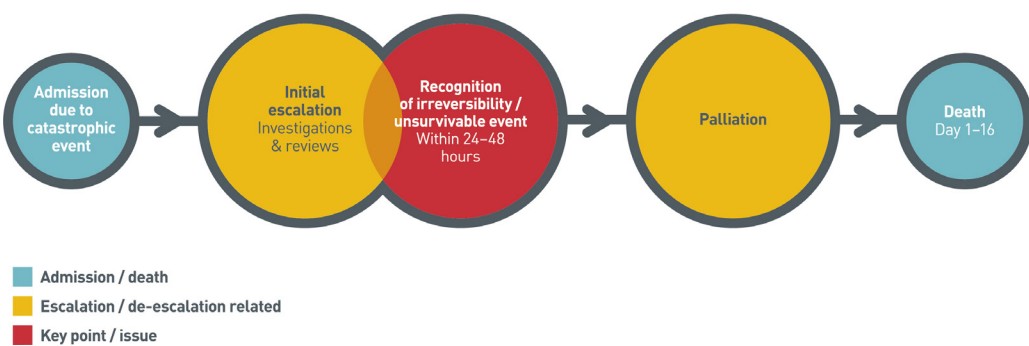

■ Admission / death
■ Escalation / de-escalation related
■ Key point / issue

**Figure 1** Early de-escalation due to catastrophic event.

(figure 4). The triggers for setting limits included patient refusal of treatment, discussions with family, senior clinician review and marked deterioration in the patients' condition. Crucially, these triggers occurred against backgrounds of: history of recent deterioration, frailty, underlying diagnoses of dementia or malignancy and the presence of pre-existing DNACPR orders and palliative care involvement. In line with this, the patients in this trajectory had the highest average comorbidity scores and ages.

Early treatment limits formed the start of a multistaged de-escalation process, which occurred across the duration of admission. This de-escalation started with treatment limits (DNACPR, not for intubation/dialysis/ICU care and ward-based care) before progressing to more active de-escalation (ceasing early warning scores, ceasing antibiotics/intravenous fluids/regular medications, palliation and commencement of an individualised end of life care plan).

Key to this trajectory was the level of ongoing treatment in parallel with the staged de-escalation. Despite early treatment limits being set, ongoing treatment involved

a far more extensive range of treatment (interventions, therapy and medications) than in trajectory 3. Interventions included catheterisation, nasogastric tubes and blood transfusions. There was therapy input from physiotherapists, speech and language therapists, dietitians and occupational therapy teams, and medications included diuretics and antibiotics. Nonetheless, ongoing treatment was restricted to a ward environment as clinical history meant these individuals were not candidates for intensive treatment.

### The categories

In addition to the care management typology, our directed content analysis revealed a number of contextual issues, which influenced decision making. We identified three interlinked categories consisting of: multiple clinician involvement, family involvement and lack of planning clarity. The categories were framed by clinical complexity and uncertainty.

### Clinical complexity and uncertainty

The cases demonstrated clinical complexity caused in the main by multiple comorbidities, new diagnoses or

CARE MANAGEMENT TRAJECTORY TYPE 2

**TREATMENT WITH CURATIVE INTENT THROUGHOUT**

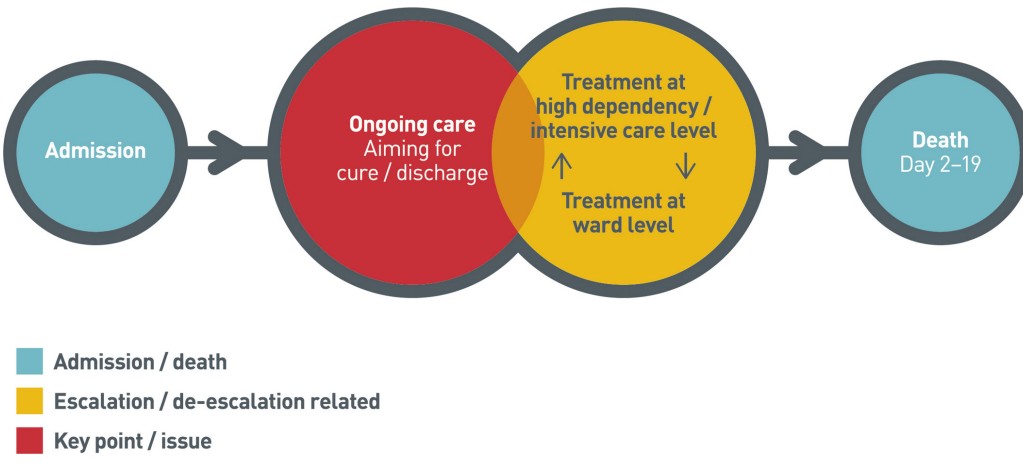

■ Admission / death
■ Escalation / de-escalation related
■ Key point / issue

**Figure 2** Treatment with curative intent throughout.

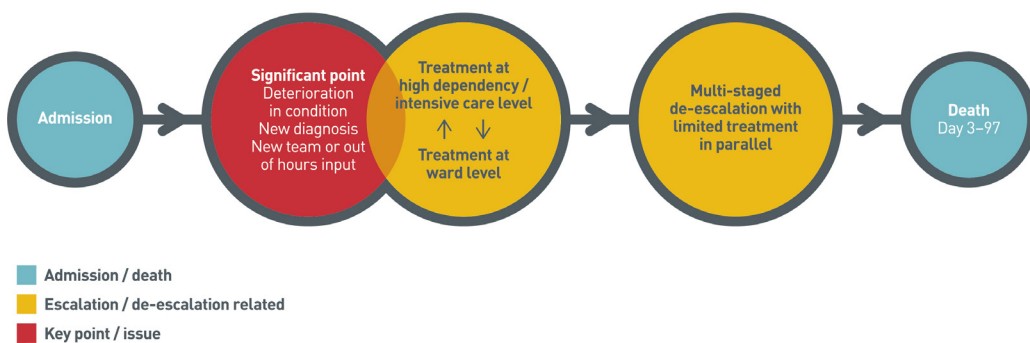

CARE MANAGEMENT TRAJECTORY TYPE 3

**TREATMENT WITH CURATIVE INTENT UNTIL SIGNIFICANT POINT**

**Figure 3** Treatment with curative intent until significant point.

undiagnosed conditions and challenging management, for example, of sepsis, kidney injury and frailty. Challenging management of fluid balance issues associated with multiple concurrent comorbidities, and the onset of new infections, were a frequent occurrence. A lack of clarity surrounding definitive diagnoses often meant that clinicians were 'working in the dark' trying to maximise management despite ongoing uncertainty. Although there were some more clearly defined diagnoses and management paths evidenced (such as stroke), with greater clinical predictability, these cases were in the minority.

Decision making was complicated by frequently changing clinical situations, particularly in relation to new findings or diagnoses. Escalation-related decisions were required that could adapt to these changing situations, where previous management plans were rapidly rendered inappropriate.

## Multiple clinician involvement

Clinical management via multiple specialities, therapy and outreach teams could preclude sight of the patient's prognosis. This was evidenced by treatment decisions and therapy involvement that did not always reflect an individual's prognosis. Likewise, the practicalities of input from multiple specialisms, including numerous repeat reviews and interplay between different teams, often acted to elongate decision-making processes, and added complexity when no-one team took responsibility for leading decisions.

There was evidence of a hierarchy in decision making, with senior clinicians most often instigating decisions. Junior doctors were less likely to make escalation-related decisions, especially concerning placing limitations on, or removal of, treatments. Junior doctors, when required to make decisions alone (particularly those working out of hours), were more likely to continue treatment escalation, especially in the absence of prespecified escalation plans. As such, there was a clear role for senior review, with registrars and consultants instigating the majority of decisions regarding treatment limits and withdrawal of treatment.

The transfer of patients between wards and clinical teams added complexity to decision processes. There was evidence of transfers resulting in de-escalation plans being overlooked; however, in other circumstances, ward or team moves prompted new reviews and the initiation of appropriate planning. The positive influence of new perspectives or 'fresh eyes' on escalation-related decision making was apparent, especially via out-of-hours clinicians. It appeared that individuals not caught up

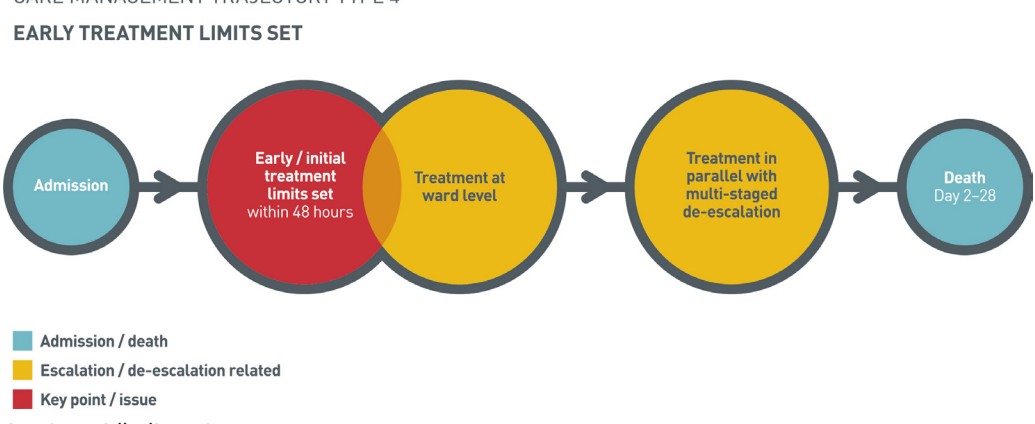

CARE MANAGEMENT TRAJECTORY TYPE 4

**EARLY TREATMENT LIMITS SET**

**Figure 4** Early treatment limits set.

in the day-to-day management of patient care were able to see the 'bigger picture' regarding care management, often initiating ceilings of care, or prompting escalation plans.

### Family involvement

The role and influence of the family was often central in the decision-making process. It was apparent that escalation-related decisions (ie, whether to continue to increase the intensiveness of treatment, for example, dialysis, intubation and ventilation or maintain treatment at ward level) were often established and actioned before discussions with the family took place, whereas, de-escalation-related decisions (such as ceasing treatments and commencing palliation) were postponed until after discussions with family. Family involvement and consensus agreement always preceded the withdrawal of treatment (eg, organ support and ventilation). This reflects the moral imperative to discuss such decisions with family. Additionally, family were involved in the decision-making process for DNACPR orders where there was any concern about patient competency. Families also played an important role in providing collateral histories for clinicians, enabling decision making to be placed in the context of an individual's recent health. This was particularly the case with older patients where families could highlight weeks or months of recent deterioration or recurrent infections, aiding the admission clerking, and facilitating early treatment limits being set (trajectory 4).

The impact of the familial role was most apparent when absent. In a few cases, where patients had limited or no family involvement, or lacked the physical presence of family members to prompt discussions, de-escalation decisions were not made (those in trajectory 2). In contrast, where families were engaged, they were frequently involved in consultative decision making with clinical teams. These families were often able to provide clear instructions to clinicians because of their knowledge of patients' prior wishes. For example, relatives were recorded as stating that the '*patient wouldn't want to live like this*' and were therefore more likely to endorse clinician recommendations for treatment withdrawal. Additionally, families often agreed with recommendations that if the patient did not respond to treatment, then a move to focus on palliation should occur. The converse did apply, although only in a few cases, whereby families stated that the patient would '*want all done*'. In situations where families were unsure of the patient's wishes, further team meetings with the family were always undertaken.

### Lack of planning clarity

The data revealed a general lack of clarity and visibility regarding management plans in the case notes. However, the clinical complexity of these cases at times precluded the making of escalation-related plans or led to them being held in a type of uncommitted management 'status' until certainty was gained. Even where cogent management plans were made, they may not have been followed because there were no effective methods for signposting clinicians to plans buried in subsequent pages of notes. In addition, where management plans involved clear de-escalation, these were not always followed. This was sometimes more than just due to the lack of visibility in the notes and also due to clinical complexity and unpredictability of deterioration, with fluctuations leading to patients temporarily improving or stabilising.

Initial clerking and history taking was paramount to the quality of decision making throughout admission. This was particularly apparent where clerking histories appeared 'lost', with key factors not carrying through into decisions made. Where an important comorbidity was not acknowledged during the admission clerking, this could continue to influence care over the length of admission.

### DISCUSSION

This case note review and qualitative analysis, identified four care management trajectories, defining and mapping clinical decision-making processes in the context of acute pathophysiological deterioration. All trajectories from admission through to death were framed by clinical complexity and related uncertainty. In general, such complexity confounded decision-making processes. Nonetheless, in a minority of profoundly complex cases (eg, older age, associated frailty and comorbid and premorbid statuses), complexity could encourage escalation-related decision making. This was apparent in the fourth trajectory, where early treatment limits were set based on patients' preadmission morbidity. This concurs with Fritz et al[14] who in a retrospective case note review found a lower threshold for completing DNACPR orders in patients with multiple co-morbidities.

The trajectories identified here expand those previously described by Higginson et al, which were exclusive to critical care, as they are applicable to hospital inpatients irrespective of care setting.[4] Consequently, our trajectories highlight: (a) significant points in care trajectories where senior secondary review and re-evaluation of management plans would be valuable and (b) groups of patients for whom a formal TEP would be of particular benefit, as a framework to support discussions and the recording of decisions.

Our findings display significant points in care management trajectories (1 and 3). These included the recognition of irreversibility, deterioration in current condition, new diagnoses leading to deterioration and new clinical or out-of-hours team involvement. It was these points that triggered discussions around escalation and ultimately decision making. We propose that while acting as triggers, these points in trajectory 3 cases also present missed opportunities, for earlier, timely decision making. It was frequent for deterioration to occur out of hours, with important decisions left to on-call teams and sometimes more junior clinicians. As previous studies have shown, this can preclude decisions that reflect the best interests

and preferences of the patient.[12] Here, clear management plans are required that pre-empt the possibility of deterioration and outline the patients' wishes in such circumstances, as well as realistic parameters of care.

The absence of significant points in some cases by which to trigger decision making, such as those in trajectory 2, leads to a proposition made by the study's expert clinical reviewers that strategic senior reviews are required. It is possible that earlier senior review secondary to a postadmission review may enable appropriate re-evaluation and alter management plans. Nevertheless, a lack of recognition of the dying phase, even by senior clinicians, highlighted the role and contribution of palliative care teams in questioning ongoing investigations or treatment and stimulating appropriate symptom control.

It is known that formal TEPs are helpful in stimulating discussions, formulating clear plans, ensuring patient preferences are considered[5 15] and perceived as a good idea by patients, families and healthcare professionals.[5 15–17] In addition, they help healthcare professionals structure their discussions with patients and families, and record their decisions, improving documentation clarity[18] and escalation-related communication within clinical teams.[16 19] Despite this, in the case notes reviewed, there were no recorded instances of a formal TEP being used to aid decision making. Four patients held pre-existing DNACPR orders, but none had evidence of an advance care plan or formal TEP. Despite the small number of pre-existing DNACPR orders in the review, their existence led clinicians to have wider escalation-related discussions with patients and families. There is also a pragmatic argument that documenting a DNACPR decision should trigger consideration of a TEP, as a logical continuation of the resuscitation discussion. However, based on our care trajectories, treatment escalation decision making must account for premorbid status, which may, if possible, be best assessed outside of crisis situations and acute deterioration. To incorporate patient preferences, completion of formal TEPs in primary care would enable patients who might be too acutely unwell on admission to hospital to participate in such discussions (of particular relevance to trajectories 1 and 4). Although it is impossible to anticipate the catastrophic events that occurred for individuals in trajectory 1, it is contended that those individuals who have significant comorbidities and resulting premorbid dependencies (such as those in trajectory 4) should be party to sensitive discussion and documentation of a TEP in primary and community care settings.

In summary, this review has highlighted a number of clinically relevant findings, with resulting recommendations, which the authors contend might represent best practice:

► Accurate history taking surrounding premorbid functional status, comorbidity and level of dependency is vital for establishing ceilings of care.
► Regular senior clinician involvement results in ongoing review of prognosis and facilitates effective decision making in complex patients where there is significant clinical uncertainty.
► Awareness of a patient's premorbid wishes and, where possible, discussion with the patient, should be a priority in deciding ceilings of care.
► Discussion with family around prognosis should complement discussions with the patient.
► 'Fresh eyes' are a valuable tool for reassessing patients' prognosis and should be used more widely for complex patients with significant clinical uncertainty, not responding to treatment.
► A senior clinician with overall responsibility for the patient should facilitate multidisciplinary discussion of patients with multiple team involvement.
► Earlier involvement of palliative care specialists in patient assessment would aid decision making and recognition of those who are at the end of life.
► Formal TEPs do not preclude active management of reversible conditions but would aid decision making and need to be introduced and adopted by clinical teams.
► Patients with TEPs need these to be readily visible to teams providing ongoing care to ensure they are followed.

## CONCLUSION

This review highlighted the complex care management and related decision-making processes of individuals who face acute pathophysiological deterioration leading to death in hospital. Such decision-making processes involve multiple layers of clinicians, from numerous specialities, within often hierarchical teams. Families were involved in contributing to decision making; in these circumstances, patients themselves were frequently too acutely unwell to contribute to all stages of the process. The review identified the need for visibility and clarity of management plans, in spite of the surrounding frame of clinical uncertainty. Even where clear plans were documented they could be buried by subsequent pages of notes, with no effective signposting, a particular problem when further deterioration occurred out of hours. Therefore, the review suggests that there is a clear role for formal TEPs to be introduced more widely into routine practice. Opportunities need to be created for patients and their families to be able to ask for such plans to be made, in consultation with clinicians who know them best, outside of the circumstances of acute deterioration.

**Acknowledgements** The authors would like to thank Dr Marion Penn for her assistance with stratification of the sample. The authors would also like to thank Mrs Sally Dace for her patient and public focused involvement and the expert clinicians who reviewed the analysis and contributed to the overall interpretation of the data.

**Contributors** All authors designed the review. NC applied for the necessary approvals. NC and AC extracted the data. NC and AC analysed the data with assistance from SL, NWP and AR. NC and AC drafted the manuscript with assistance from SL, MM, CRM, NWP and AR. All authors critically reviewed the manuscript for intellectual content and approved the final version of the paper.

**Funding** This work was supported by the National Institute for Health Research Collaboration for Leadership in Applied Health Research and Care (NIHR CLAHRC)

Wessex, which is a partnership between Wessex NHS organisations and partners, including the NIHR Southampton Biomedical Research Centre, and the University of Southampton.

**Disclaimer** Funders had no role in study design, data collection and analysis, decision to publish or preparation of the manuscript. The views expressed are those of the author(s) and not necessarily those of the NHS, the NIHR or the Department of Health and Social Care.

**Competing interests** None declared.

**Patient consent** Not required.

**Ethics approval** Health Research Authority and Research Ethics Committee South Central – Hampshire A.

**Provenance and peer review** Not commissioned; externally peer reviewed.

**Data sharing statement** The datasets generated and analysed during this study are not available due to the nature of approval for the study. Support under section 251 of the NHS Act (2006) was obtained via the Health Research Authority's Confidentiality Advisory Group as access to patient identifiable data was required without consent; therefore, no additional data can be made available.

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
