## [Reviewer comments · BMJ Open]

ARTICLE DETAILS

TITLE (PROVISIONAL)	ESCALATION RELATED DECISION-MAKING IN ACUTE DETERIORATION: A RETROSPECTIVE CASE NOTE REVIEW
AUTHORS	Campling, Natasha; Cummings, Amanda; Myall, Michelle; Lund, Susi; May, Carl; Pearce, Neil; Richardson, Alison

VERSION 1 – REVIEW

REVIEWER	Linda Ozekcin. Lead Clinical Nurse Specialist for Critical Care Division St. Luke's University Health Network, Bethlehem, PA, USA
REVIEW RETURNED	26-Feb-2018

GENERAL COMMENTS	In the conclusion, the reference to individuals, is this indicating that patients and their families request formal TEPs? Perhaps adding suggestions for how to create these opportunities especially in a period of clinical uncertainty and ways to measure increased satisfaction with the health care experience.
---

REVIEWER	Dr Mike Charlesworth Wythenshawe Hospital, Manchester, UK
REVIEW RETURNED	17-Apr-2018

GENERAL COMMENTS	This is an important topic and I agree that research in this area is lacking. It is also a difficult area to study and I agree that a qualitative approach is appropriate. I think the themes are clinically relevant and interesting. I agree that formal TEPs would greatly enhance patient safety, prevent unnecessary interventions and improve our ability to deliver sustainable healthcare. I would be interested to know what barriers there are to their introduction, though I appreciate this would require a different study. I assume there would be many! Minor points - the introduction is too long and I'm not sure what value table 1 adds. Some of the findings are not that striking. For example, yes we do need to take accurate histories about functional status, but despite most generally accepting this is important, it is often done poorly. Senior clinicians should be regularly involved with their patients, but they also have many other responsibilities too. Patient/family wishes should always be taken into account. We already know this, but these points are the main clinically relevant findings presented. What, therefore, does this study tell us that is new? As an aside, I like the pictorial representation of case management
--

	trajectory types. From my own practice, I think they are clinically accurate.
--	---

VERSION 1 – AUTHOR RESPONSE

Reviewer 1 Comments:

In the conclusion, the reference to individuals, is this indicating that patients and their families request formal TEPs? Perhaps adding suggestions for how to create these opportunities especially in a period of clinical uncertainty and ways to measure increased satisfaction with the health care experience.

Thank you for these points of clarification. Yes the reference to individuals does refer to patients and their families, we have altered this in both the abstract and conclusion to ensure this is now clear. We argue that opportunities to request and document a formal treatment escalation plan need to be created prior to situations of acute deterioration. We have re-worded the respective sentences to refer to situations “outside of the traumatic circumstances of acute deterioration”.

Reviewer 2 Comments:

This is an important topic and I agree that research in this area is lacking. It is also a difficult area to study and I agree that a qualitative approach is appropriate. I think the themes are clinically relevant and interesting.

Thank you we agree this is an important topic, lacking in research and is a current gap in knowledge.

I agree that formal TEPs would greatly enhance patient safety, prevent unnecessary interventions and improve our ability to deliver sustainable healthcare. I would be interested to know what barriers there are to their introduction, though I appreciate this would require a different study. I assume there would be many!

In the background to the paper we refer to the wider programme of work on TEPs being undertaken by the authors, examining the implementation and evaluation of TEPs including the barriers to their introduction. This paper reports findings from a study which compromised one of the work packages from this programme.

Minor points - the introduction is too long and I'm not sure what value table 1 adds.

Re: Manuscript ID bmjopen-2018-022021 entitled "ESCALATION RELATED DECISION-MAKING IN ACUTE DETERIORATION: A RETROSPECTIVE CASE NOTE REVIEW"

Thank you for considering this paper for publication in BMJ Open. We have responded to the reviewers' comments as below and made changes to the manuscript (highlighted using track changes). We would like to thank the reviewers for their very helpful comments.

Reviewer 1 Comments:

In the conclusion, the reference to individuals, is this indicating that patients and their families request formal TEPs? Perhaps adding suggestions for how to create these opportunities especially in a period of clinical uncertainty and ways to measure increased satisfaction with the health care experience.

Thank you for these points of clarification. Yes the reference to individuals does refer to patients and their families, we have altered this in both the abstract and conclusion to ensure this is now clear. We argue that opportunities to request and document a formal treatment escalation plan need to be created prior to situations of acute deterioration. We have re-worded the respective sentences to refer to situations “outside of the traumatic circumstances of acute deterioration”.

Reviewer 2 Comments:

This is an important topic and I agree that research in this area is lacking. It is also a difficult area to study and I agree that a qualitative approach is appropriate. I think the themes are clinically relevant and interesting.

Thank you we agree this is an important topic, lacking in research and is a current gap in knowledge.

I agree that formal TEPs would greatly enhance patient safety, prevent unnecessary interventions and improve our ability to deliver sustainable healthcare. I would be interested to know what barriers there are to their introduction, though I appreciate this would require a different study. I assume there would be many!

In the background to the paper we refer to the wider programme of work on TEPs being undertaken by the authors, examining the implementation and evaluation of TEPs including the barriers to their introduction. This paper reports findings from a study which compromised one of the work packages from this programme.

Minor points - the introduction is too long and I'm not sure what value table 1 adds.

We consider that the background to the paper fits the journal requirements for submission and frames what this study adds to the literature. For this reason we have not edited it further.

The authors believe that Table 1 is key and does add value to the paper. It outlines the key characteristics of the patients within the overall sample, and those categorised to each of the four trajectories. The table includes at the individual level the reasons for the "key point/issue" in each trajectory: triggers for recognition of irreversibility (trajectory 1); reasons for on-going care/no de-escalation (trajectory 2); significant point triggering de-escalation (trajectory 3); and prompts for early treatment limits (trajectory 4). This level of detail is not included elsewhere in the paper.

Some of the findings are not that striking. For example, yes we do need to take accurate histories about functional status, but despite most generally accepting this is important, it is often done poorly. Senior clinicians should be regularly involved with their patients, but they also have many other responsibilities too. Patient/family wishes should always be taken into account. We already know this, but these points are the main clinically relevant findings presented.

What, therefore, does this study tell us that is new?

Thank you for highlighting this. We have added an additional point to the section outlining the strengths of this study. The first 2 bullet points address what is new:

- There is a lack of description of escalation-related decision-making in the context of deterioration outside of the critical care environment. Our study setting was the comprehensive hospital environment, and individuals who were facing acute deterioration that led to death.*
- The study explored clinical decision-making processes: the types and range of decisions made, the involvement of families in these processes, and the interaction between clinical teams. Care management trajectories provoked by acute deterioration were characterised, via typology; including points of significance in the sequence of events. Contextual issues influencing decision-making were described: multiple clinician involvement, family involvement and lack of planning clarity; all framed by clinical complexity and uncertainty.*

Thank you again for considering our paper for publication in your journal. We hope you feel we have responded adequately to the reviewers' comments and look forward to hearing from you in due course.

VERSION 2 – REVIEW

REVIEWER	Mike Charlesworth Manchester University NHS Foundation Trust, UK
REVIEW RETURNED	06-May-2018
GENERAL COMMENTS	Thank you for your comments and revisions. I am now satisfied that

	this manuscript can be accepted for publication.
REVIEWER	Linda Ozekcin DNP RN CCRN CCNS St. Luke's University Health Network, USA
REVIEW RETURNED	30-May-2018
GENERAL COMMENTS	Thank you for your time in revising the manuscript! The topic is very relevant to the clinical scenarios that health care professionals deal with often on a daily basis to provide guidance to patients and their families in making difficult decisions.